# Clinical Application of Personalized Digital Surgical Planning and Precise Execution for Severe and Complex Adult Spinal Deformity Correction Utilizing 3D Printing Techniques

**DOI:** 10.3390/jpm13040602

**Published:** 2023-03-30

**Authors:** Hongtao Ding, Yong Hai, Lijin Zhou, Yuzeng Liu, Yiqi Zhang, Chaofan Han, Yangpu Zhang

**Affiliations:** Department of Spine Surgery, Beijing Chao-Yang Hospital, Capital Medical University of China, Beijing 100020, China

**Keywords:** spine deformity, osteotomy guidance, three-dimensional printing, personalized design, digital surgery simulation

## Abstract

(1) Background: The three-dimensional printing (3DP) technique has been reported to be of great utility in spine surgery. The purpose of this study is to report the clinical application of personalized preoperative digital planning and a 3DP guidance template in the treatment of severe and complex adult spinal deformity. (2) Methods: eight adult patients with severe rigid kyphoscoliosis were given personalized surgical simulation based on the preoperative radiological data. Guidance templates for screw insertion and osteotomy were designed and manufactured according to the planning protocol and used during the correction surgery. The perioperative, and radiological parameters and complications, including surgery duration, estimated blood loss, pre- and post-operative cobb angle, trunk balance, and precision of osteotomy operation with screw implantation were collected retrospectively and analyzed to evaluate the clinical efficacy and safety of this technique. (3) Results: Of the eight patients, the primary pathology of scoliosis included two adult idiopathic scoliosis (ADIS), four congenital scoliosis (CS), one ankylosing spondylitis (AS), and one tuberculosis (TB). Two patients had a previous history of spinal surgery. Three pedicle subtraction osteotomies (PSOs) and five vertebral column resection (VCR) osteotomies were successfully performed with the application of the guide templates. The main cobb angle was corrected from 99.33° to 34.17°, and the kyphosis was corrected from 110.00° to 42.00°. The ratio of osteotomy execution and simulation was 97.02%. In the cohort, the average screw accuracy was 93.04%. (4) Conclusions: The clinical application of personalized digital surgical planning and precise execution via 3D printing guidance templates in the treatment of severe adult rigid deformity is feasible, effective, and easily generalizable. The preoperative osteotomy simulation was executed with high precision, utilizing personalized designed guidance templates. This technique can be used to reduce the surgical risk and difficulty of screw placement and high-level osteotomy.

## 1. Introduction

Severe and complex adult deformities have characteristics of anatomical complexity, rigidity, and severe kyphosis or scoliosis, with or without neurological dysfunction, which may lead to physical disability or even death. The treatment strategy for such severe deformities requires comprehensive consideration, not only due to the high magnitude of spinal deformities and the low compliance of the spine, but also the possible presence of spontaneous or iatrogenic fusion. How to comprehensively evaluate the characteristics of spinal deformities in the perioperative period and to develop personalized and precise surgical treatment regimens based on their characteristics are the key factors affecting correction outcomes.

With the development and rapid application of the three-column fixation technique for different kinds of spinal osteotomies, spinal deformity and correction currently achieve satisfactory correction results. It is important to note that the complex spinal structures described above can pose significant challenges in screw insertion, vertebrae osteotomy, correction, and decompression procedures. These difficulties can lead to a higher incidence of intra- or postoperative neurological complications, which have been reported to occur as frequently as 4–23% [1,2,3,4,5]. According to Chen et al., the risk is closely related to preoperative severity of deformity, intraoperative osteotomy site, osteotomy type, and kyphosis correction rate [6]. The preoperative severity of deformity is uncontrolled, and how to ensure accurate implantation of internal fixation and to design and perform an optimal osteotomy at an optimal position is crucial to the success of the procedure.

Personalized three-dimensional (3D) surgical simulation is reported to have been applied in spine deformity surgery in order to help the surgeon obtain an optimal spinal alignment. Reliability in real spine surgery and in simulated 3D spine models of the computer-assisted technique in posterior osteotomies of thoracolumbar kyphosis has been verified in previous works [7]. This technique is believed to provide intuitive reference to, and comprehension of, posterior osteotomy, and to aid surgeons’ anatomy measurement and correction design preoperatively in spinal deformities.

The three-dimensional printing (3DP) technique has been verified to be of great utility in spine surgery, including preoperative planning, anatomic visualization, custom prosthetic design, and even as an educational tool for training and doctor–patient communication [8,9,10]. In addition, for spine deformity patients, it can offer assistance in pedicle insertion and osteotomy procedures. It has been demonstrated to benefit patients in improving pedicle screw insertion accuracy, shortening surgery duration, decreasing blood loss and preventing complications [11,12,13,14]. Preoperative planning and pedicle subtraction osteotomy (PSO) assisted by 3DP template has been reported by Xin et al., who presented the process of osteotomy guide plate construction and simulated surgery, demonstrating that the 3DP PSO guide plate system can be used for preoperative osteotomy planning with good accuracy [15]. However, their study was executed in vitro with several limitations, which discount the value of this novel personalized surgery system.

Hence, to obtain an optimal surgical outcome, we combined personalized digital planning and 3DP technique assisted osteotomy in the correction surgery of severe and complex adult spinal deformity. Here, we introduce this technique and report the outcomes of eight patients with severe and complex adult spinal deformity.

## 2. Materials and Methods

This study was approved by the Institutional Review Board of the BJCY Hospital, CMU. Eight patients diagnosed with severe and complex adult spinal kyphoscoliosis including congenital scoliosis (CS), ankylosing spondylitis (AS), tuberculosis (TB) and revision scoliosis were enrolled retrospectively in the study after providing written informed consent. Severe kyphoscoliosis was defined as kypho/scoliosis cobb angle exceeding 80°, adult scoliosis as patient aged more than 18 years old, and rigid scoliosis as flexibility of less than 25%, according to cobb angle.

The preoperative radiological examination, including full-spine standing X-ray, bending X-ray, computed tomography (CT) scan and magnetic resonance imaging (MRI), was taken after admission. The CT scan data were used for surgery planning and guide-template construction. Thin-slice spiral CT (Siemens CT machine, SOMATOM Sensation 16, Siemens AG, Forchheim, Germany) scan data of eight patients were downloaded in the DICOM format. Mimics 17 software (Materialise, Leuven, Belgium) was used for full-spine 3D reconstruction, and the data were exported in the STL format to print or other procedures.

In Mimics software, the whole spine 3D reconstruction is present in detail and can be rotated on any axis and by any angle, therefore the characteristics of spinal deformity can be observed from different perspectives. The patient-specific assisted tools include an osteotomy-guiding template used in deformity correction and a pedicle insertion guiding template assisting screw implantation.

The principle of simulated correction surgery is determined as follows: (1) to minimize the scoliosis and kyphosis deformity; (2) to make the C7 plumb line overlap with the central sacral vertical line in the coronal position; (3) to restore the sagittal alignment, achieving a satisfactory chine brow vertical angle and recovering the SVA within 4 cm.

### 2.1. Construction of Pedicle Screw Insertion Guide Template

The DICOM data of the thin-slice CT scan (Siemens CT machine, SOMATOM Sensation 16, Siemens AG, Forchheim, Germany) of the patient was downloaded. Mimics 17 software (Materialise, Belgium) was applied for full spine 3D reconstruction, and the data was exported in the STL format to print or next procedure.

In Mimics 17, based on the repaired Mask, the Morphology operations command, Boolean operation command and Part command were applied to establish the inflated part of the lamina.

The repaired Mask was applied to construct the model, the Cylinder command was used to establish the screw trajectory, and the screw trajectory guide pin was adjusted to the optimal position via the three perspective views (Figure 1a). According to the surgical requirements, the design of the screw trajectory of the instrumented segment was carried out, performing the Cut Orthogonal to Screen command and the Boolean command to precisely cut out the unnecessary expansion mask, in order to obtain a single or several vertebral body fitting guides (usually no more than 3 vertebral bodies). Then the Boolean command was used to remove the excess part of the coupling surface between the guide plate and the bone, so that the guide plate can fit the cortical bone perfectly.

The fixed connecting rod of the guide plate was drawn in the ProE (Parametric Technology Corporation, Boston, MA, USA) software (adding annotations to the connecting rod to identify the guide plate), and the drawn connecting rod transferred to Mimics. The Boolean command was used to fuse the plates with the connecting rods. Expanding the diameter and length of the screw guide pin model to obtain the profile of the screw guide, the Boolean command was applied to integrate the trajectory with the guide plate. Reducing the diameter of the screw guide pin, the Boolean command was used to subtract the space of the pin to obtain the complete screw track. The design of the screw guide plate was complete (Figure 1b).

### 2.2. Construction of Osteotomy Guide Template

The grade, range and position of the osteotomy are determined according to the correction surgery needs (Figure 2a,b). The operation can be repeated until a satisfactory correction outcome is obtained, and the plane of the osteotomy is determined (Figure 3). Measuring the length of the guide plates and drawing the guide rails in the ProE (adjusting the width according to the size of the ultrasonic osteotome, generally a width of +0.4 mm), then transferring it into Mimics for position calibration to make the osteotomy guide rail overlap with the osteotomy plane (Figure 2c). After calibration, the Cut Orthogonal to Screen command was used to cut the appropriate length (the length should be less than the ultrasonic osteotome, to make sure the cutting depth of the osteotome is sufficient and safe, which means that the ultrasonic osteotome can cut the lamina but will not injure the spinal cord), and the Boolean command was employed to integrate it with the guide plate (Figure 2d).

### 2.3. Model and Template Printing

Once the above-described steps were completed, the surgical plan was finished. The designed osteotomy guiding template and the pedicle positioning and guiding template in the software can be transferred to the 3D printer to manufacture the entitative model and templates. The spine model and guiding templates were printed at the original 1:1 ratio by 3D print technology. Resin was used as the printing material. The print error of the model was <0.2% per 10 cm. The printing templates were delivered to the operation room to be sterilized and packaged, and are supplied to the surgeon when the surgery is performed (Figure 4).

### 2.4. Surgical Procedure

After general anesthesia, the patient was fixed firmly prone on the surgery table. Subperiosteal exposure of the posterior elements was performed from upper instrumented vertebra to lower instrumented vertebra through a posterior middle line incision, to make sure the lamina and facet joint were clearly exposed (Figure 5a). The 3DP screw insertion guide template was placed on the surface of the lamina and spinous processes, while ensuring firm bone contact and a perfect match for the 3DP guide template (Figure 5b). The Kirschner wires were placed into the pedicle through the foraminule in the guide template, then the template and Kirschner wires were removed, and the accuracy and completeness of the pedicle screw trajectory is checked, before the screw is inserted (Figure 5c–f). What should be noted is that the four screws adjacent to the osteotomy segment should be placed after the osteotomy rails are drawn, because the screw insertion guidance of the adjacent segment and the osteotomy guidance template were manufactured integrally to help fix the template.

After the screw insertion is completed, the osteotomy is performed. The spinous processes are first removed, if necessary. Then the osteotomy guide template is placed to ensure a perfect match (Figure 5g). The Kirschner wires are inserted into adjacent pedicles of the non-osteotomy vertebrae to fix the template, using the ultrasonic osteotome to cut the lamina along the guide rail of the template, then the template is removed (Figure 5h,i). A temporary rod contoured to the shape of the deformity is always fixed first, in order to maintain a stable structure before the resection procedure. The spinous process, facet joint, total laminectomy, transverse process and the rib on the resection side are first resected, with the osteotomy rails as reference. The entire vertebral body and adjacent discs are removed in pieces via the pedicle approach after pedicle resection.

In VCR osteotomy cases, an anterior fusion is performed with a titanium cage as structural support to prevent too much shortening of the spinal column. The resected gap is then gradually compressed by repeated manual compression and shortening of the convex side. The final rods are fixed after obtaining an optimal curvature, and the satellite rod and cross-link are connected if necessary (Figure 5j).

### 2.5. Data Collection

The perioperative, radiological parameters and complications were collected retrospectively, including surgery duration, estimated blood loss, pre- and post- operative cobb angle, trunk balance, screw accuracy, etc. The angle was measured in the three-dimensional simulation software, according to the preoperative CT scan reconstruction of the whole spine model. The cobb angle is the angle of the upper endplate of the upper end vertebrae and the lower endplate of the lower end vertebrae. The coronal trunk balance is the distance between the C7 plumb line and the central sacral vertical line. The sagittal trunk balance is the distance between the C7 plumb line and the vertical line of the S1 posterior edge of the upper endplate. The screw accuracy was assessed based on the postoperative CT scan. Gertzbein and Robbins standard was used to classify the screws and define the screw accuracy, including Grade A to E [16]. Grade A and B screws were defined as accurate.

The ratio of osteotomy execution and simulation (ROES, %) was described as the coronal and sagittal part, which means the coronal/sagittal cobb angle of the osteotomy segment divided by the cobb angle of the osteotomy segment (Figure 6). In addition, surgical outcomes at the last follow-up were defined as excellent, good, fair, or poor according to the modified MacNab criteria. Complications such as nerve root injury, spinal cord injury, and cerebrospinal fluid (CSF) leakage were recorded and analyzed.

## 3. Results

A total of eight patients were enrolled in the study, including four males and four females. The average age was 33.75 years old. The primary disease consisted of two ADIS, four CS, one AS and one TB, and two patients had a previous history of spinal surgery. There were six kyphoscoliosis and two kyphosis in regard to deformity characteristics. PSO and VCR osteotomies were performed for these patients.

The average surgical duration was 361.25 min, average blood loss 800 mL, and no surgery or instrument related complications occurred (Table 1). The cobb angle of the main curve was corrected from 99.33° to 34.17°, and the kyphosis was corrected from 110.00° to 42.00°. The average coronal and sagittal ROES were 98.19% and 96.02%. The average screw accuracy was 93.04% in the cohort (Table 2). All patients obtained excellent or good results based on Macnab’s criteria for clinical outcomes. The preoperative planning and surgery for case 1 and case 2 are shown in Figure 7, Figure 8 and Figure 9.

## 4. Discussion

The treatment of severe rigid adult scoliosis demands technique and experience. According to a previous work, 0.12% of patients among 12,375 of all types of spinal surgical procedure had a permanent neurologic deficit [17], and 0.26% to 1.75% of 6334 cases of adolescent idiopathic scoliosis in the Scoliosis Research Society Morbidity and Mortality database experienced neurologic complications [18]. The rate of neurologic deficit, however, increased to 8.7% in adult spinal deformity according to Kelly et al. [2]. Moreover, the neurological complication rate was reported to be as high as 17.1% in patients with more severe and complex deformities needing PVCR surgery, and 3.3% had permanent neurologic deficit [19]. Besides, in a multicenter study by Lenke et al. [20], the most common intraoperative complication was loss of spinal cord monitoring data with actual spinal cord or nerve root deficit. According to the Scoli-RISK-1 study [4], an international prospective multicenter study in 15 sites including 272 patients undergoing complex ASD surgery, 61 (23%) patients showed a decline in lower extremity motor score at discharge. Multivariable analysis revealed that older age, larger coronal deformity angular ratio, and lumbar osteotomy were the three major predictors of neurologic decline. At the same time, the abnormal anatomy of the spine in patients with congenital spinal deformities or in the presence of a history of spinal surgery further increases the difficulty of surgery. How to perform screw insertion and spinal osteotomy effectively and safely is of great importance to a successful deformity correction surgery.

With advances in computer and manufacturing technology, three-dimensional printing has been widely used in medical science, especially in orthopedic surgery. The most common application in spine surgery is visualization of the spine model for surgeons and guidance of screw insertion and osteotomy. It has been demonstrated that the customized 3DP screw drill guides and navigational templates, which are made to be anatomically accurate based on pre-operative imaging, could offer more accurate and safer screw placement during deformity correcting spine surgery [21,22]. The advantages include, but are not limited to, improvement of screw placement accuracy and precision, and reduction of operative time, perioperative complications and radiation exposure, when compared to non-3DP (fluoroscopy or CT) guided procedures [23,24,25,26].

The 3DP template assisted osteotomy procedure has proven to be practicable. Pijpker et al. [27] presented a report describing the use of osteotomy guidance, noting that the guides were useful for the initial stages of the PSO, providing a template for resection of the posterior elements in the planned asymmetric PSO, but may not work when the osteotomy has progressed to the apex, necessitating placement of stabilizing rods. Tu et al. [28] reported a cohort of severe kyphoscoliosis patients who underwent CAD-assisted simulated correction, for which pre-planned osteotomy guides and pedicle screw guides were printed in titanium. In 2021, Xin et al. [15] introduced an experiment in PSO osteotomy guide plate construction and operative simulation in vitro, and reported that the 3DP PSO guide plate system could be used for osteotomy planning and demonstrated good accuracy.

However, some limitations of previous reports discount the application value. The customized guide template based on the pre-operative image can fit the surface of the spine perfectly, especially in rigid or revision surgery, no matter the severity of deformity. The Kirschner wires for screw guidance can also help fix the osteotomy guide template, because the template was manufactured with integrity for 2–3 vertebrae. Fixing the template, inserting Kirschner wires, drawing the osteotomy trajectory, inserting the pedicle screw, the step-by-step procedure can make the osteotomy feasible and convenient in any spinal segment. In Tu’s study, the preoperative PSO osteotomy was well designed using software, and the operation was simulated repeatedly until achieving a satisfactory outcome. What should be considered is that the guide templates in the study were manufactured from titanium alloy. The cost, the fabrication equipment and the technique prevent the prevalence of meaningful and easy operation. Besides, it is a one-sided guidance for an osteotomy procedure, rather than a bilateral groove, and only the posterior column could be reached, which may decrease the consistency between preoperative design and real practice, especially in PSO and VCR osteotomy. In our practice, resin was used to print the model, which reduces the cost and shortens the manufacturing time, at the same time ensuring the production accuracy and firmness of the model, and can be sterilized at low-temperature. The bilateral groove can precisely guide the osteotome following a certain trajectory, even for the anterior and middle column in PSO and VCR osteotomy. We applied such a guide template in real spine deformity correction surgery, rather than Xin’s experiment in vitro, which proved feasible and safe, receiving a satisfying outcome. Our study exhibited a 65.60% coronal correction rate and a 61.82% sagittal plane correction, which was similar to the first published data on PVCRs (61.9% in coronal and 45.2% in sagittal) reported by Suk et al. in 2002 [29], and higher than a multicenter result of 54% coronal plane improvement and average decrease of 48° of the sagittal Cobb angle reported by Lenke et al. [20]. The ratio of osteotomy execution and simulation was as high as 97.02%. The average screw accuracy reached 93.04% in the cohort. All patients obtained excellent or good results based on Macnab’s criteria for clinical outcomes.

To correct a rigid severe scoliosis and kyphosis for patients with several fused segments, it is not easy to ensure optimal spinal alignment and complete neurological function. The complexity of the spinal structure brings challenges to surgeons, but with the help of the personalized and 3DP techniques, the situation gets better. Of all the advantages of such a technique, the most highlighted is personalized customization, and the surgical procedure can be programmed, simulated repeatedly by computer. What we can rely on in traditional correction surgery is the experience and skill of the surgeon, but we now have professional tools. Based on the patient’s general situation, characteristics of deformity and preoperative image, we can design an osteotomy type, position and range with the help of software, simulate the surgery and foresee the correction outcome. We can achieve perfect designs through 3DP technology, including accurate screw insertion and appropriate osteotomy performance. Meanwhile, it was found during our application that, for spinal deformity with the presence of spontaneous fusion and rigidity, such a technique would be more applicable, since the preoperative imaging data were not obviously correlated with the structural changes of the spine after anesthesia. These advanced techniques will provide adequate tools to reduce the difficulty of spine surgery in the near future, benefiting both patients and surgeons. In our study, under the guidance of the osteotomy template, we can achieve a better correction outcome than preoperatively designed, 97.02% of osteotomy execution based on simulation, and 93.04% screw insertion accuracy.

There are also limitations in our study. The absence of a control group and the small number of cases undermine the credibility of the conclusions [30]. The comparison results of correction rate and complication rate between guided and without guided techniques cannot be provided and analyzed by our current study. The benefits of 3DP templates need further evaluation in multicenter large sample studies. What is more, even if an average high precision osteotomy execution was achieved, three-column osteotomy, such as PSO, has not been guided precisely. The existing guide template system has unsatisfactory controls on the range of anterior and middle column osteotomy. We are working on this in clinical practice and will solve the problem in future study.

## 5. Conclusions

The clinical application of personalized digital surgical planning and precise execution via 3D printing guidance templates in the treatment of severe adult rigid deformity is feasible, effective, and easily generalizable. The preoperative osteotomy simulation is executed with high precision, utilizing personalized designed guidance templates. This can be used to reduce surgical risk and the difficulty of screw placement and high-level osteotomy.

## Figures and Tables

**Figure 1 jpm-13-00602-f001:**
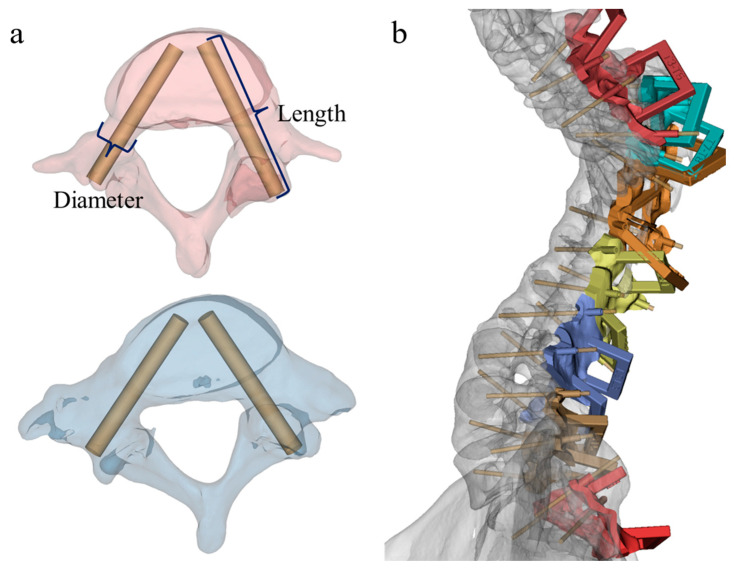
Design of screw guidance templates. (**a**) Design of screw insertion trajectory and the measurement of trajectory length and diameter; (**b**) design of screw trajectory and guidance templates.

**Figure 2 jpm-13-00602-f002:**
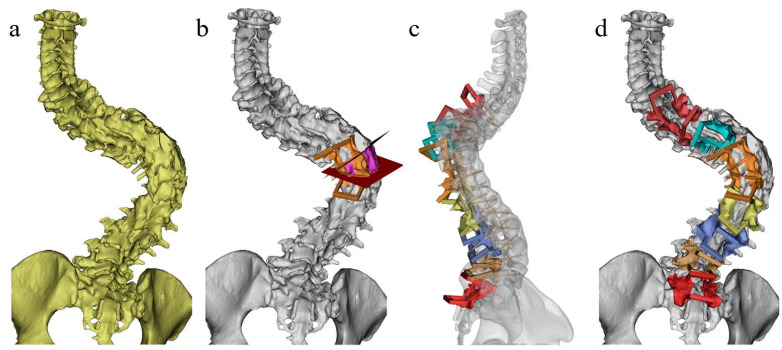
Design of osteotomy guidance templates. (**a**) Three-dimensional construction of spine; (**b**) design of osteotomy plane; (**c**) design of screw trajectory and guidance templates, making the osteotomy guide rail overlap with the osteotomy plane; (**d**) osteotomy and screw insertion guidance templates.

**Figure 3 jpm-13-00602-f003:**
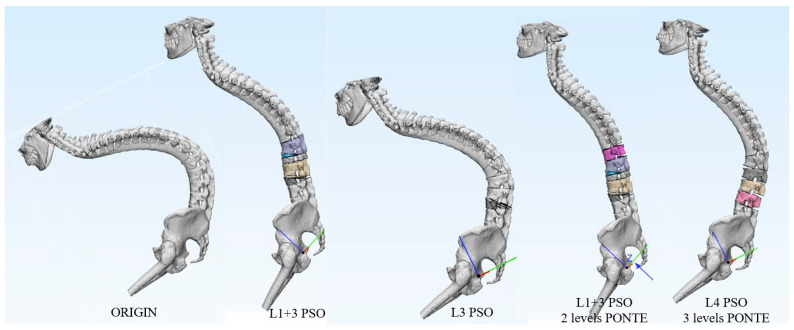
Simulation and comparation of different surgical plans.

**Figure 4 jpm-13-00602-f004:**
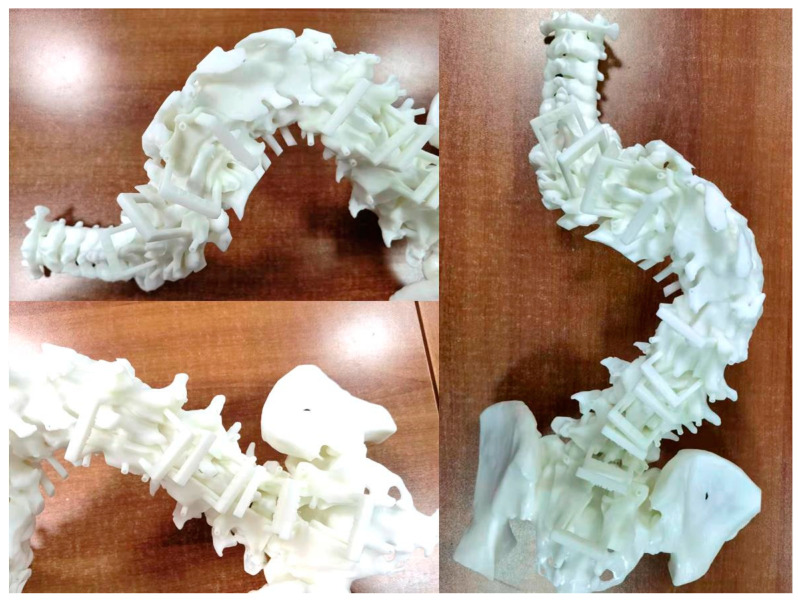
3DP spine model and guidance templates.

**Figure 5 jpm-13-00602-f005:**
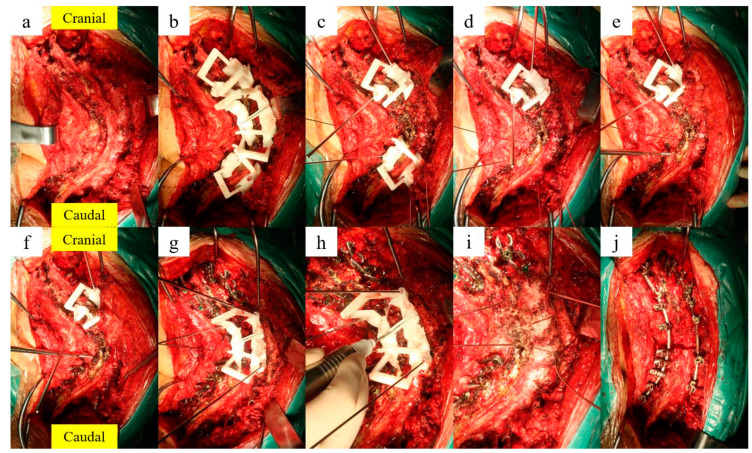
Surgical procedures step-by-step. (**a**) Exposure of spine; (**b**) the templates fit the spine perfectly; (**c**) insert Kirschner wires; (**d**) remove template; (**e**) drill the screw trajectory; (**f**) insert screws; (**g**) Kirschner wires help to fix the osteotomy guidance template; (**h**) performance of a VCR osteotomy using ultrasonic osteotome; (**i**) the osteotomy rails on the lamina; (**j**) complete correction of spine deformity.

**Figure 6 jpm-13-00602-f006:**
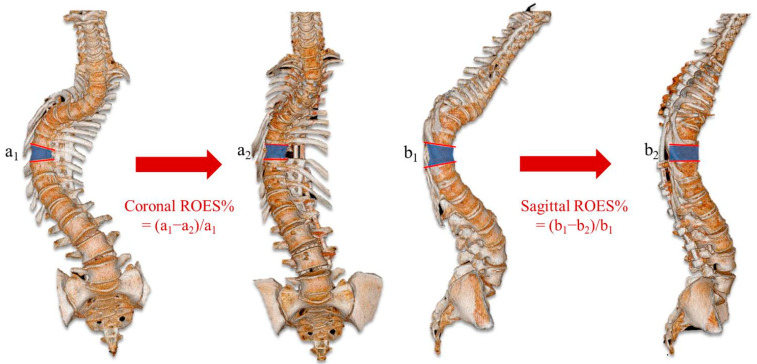
The ratio of osteotomy execution and simulation.

**Figure 7 jpm-13-00602-f007:**
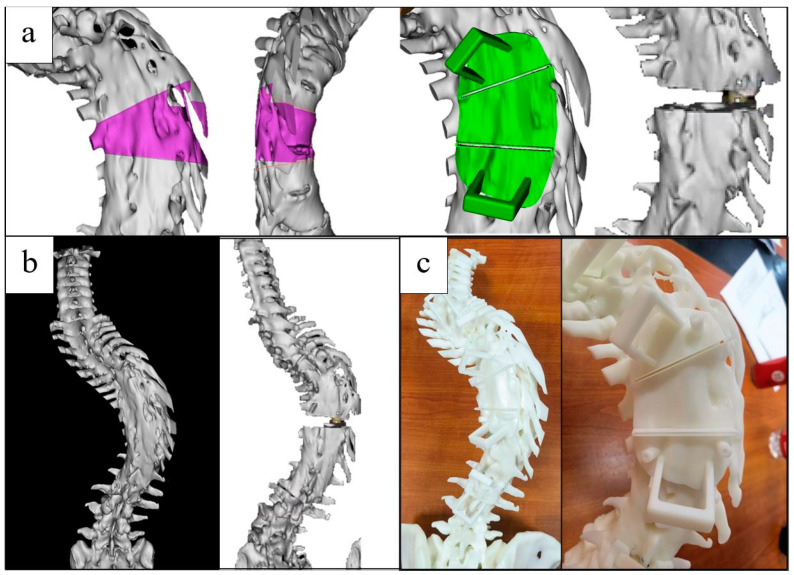
Guidance templates design for case 1 with kyphoscoliosis deformity. (**a**) Design of osteotomy and preview of outcome; (**b**) spine construction in pre- and post- designed operation; (**c**) 3DP spine model and osteotomy guidance template.

**Figure 8 jpm-13-00602-f008:**
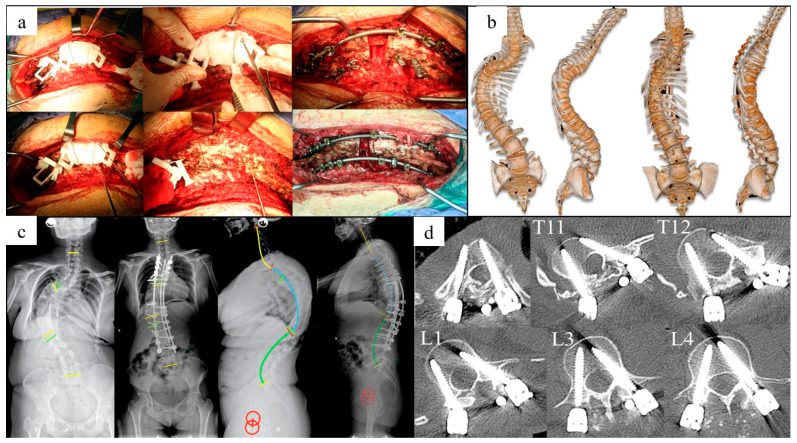
Surgical and radiological images for case 1. A 39-year-old female patient had a previous history of spinal correction surgery 26 years ago, and was given T10 VCR osteotomy and T3-L4 fixation and fusion, utilizing the personalized surgical simulation and 3DP guidance templates. (**a**) Surgery procedure of osteotomy and correction; (**b**) three-dimensional spine construction pre- and post-operation; (**c**) radiological images show great improvement of deformity between pre- and post- operation; (**d**) screw insertion accuracy in thoracic and lumbar spine.

**Figure 9 jpm-13-00602-f009:**
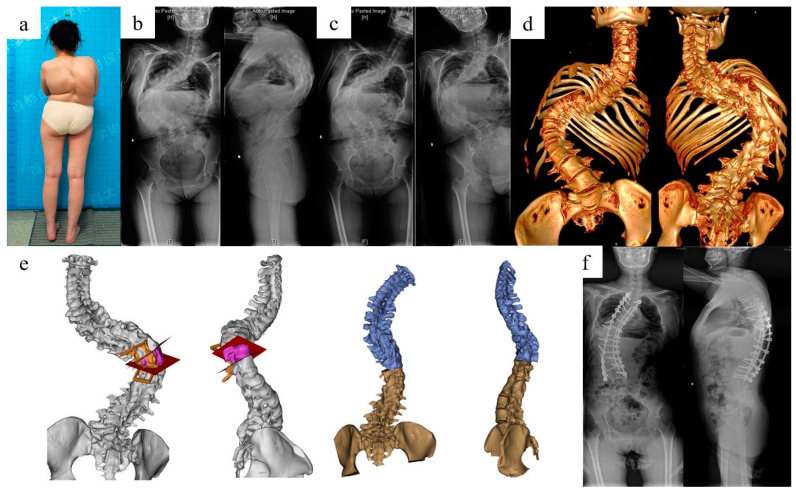
Radiological images and surgical plan simulation of case 2 with kyphoscoliosis deformity. A 33-year-old female patient had a previous history of spinal correction surgery 21 years ago, and was given T10 VCR osteotomy and T3-L4 fixation and fusion utilizing the personalized surgical simulation and 3DP guidance templates. (**a**) Outlook for case 2; (**b**) preoperative full-spine standing X-ray; (**c**) full-spine bending X-ray; (**d**) full-spine CT 3D construction; (**e**) simulation of VCR osteotomy; (**f**) postoperative full-spine standing X-ray showing great improvement for the deformity.

**Table 1 jpm-13-00602-t001:** General and operative parameters of included patients.

No.	Gender	Age (yrs.)	Diagnosis	Deformity Type	Instrumented Segments	Osteotomy Type	Osteotomy Position	Duration (min)	EBL (mL)	Blood Transfusion (mL)	Complications
1	F	39	ADIS/revision	Kyphoscoliosis	T3-L4	VCR	T10	400	700	480	N
2	F	33	ADIS/revision	Kyphoscoliosis	T3-L4	VCR	T10	420	800	800	N
3	M	43	CS	Kyphoscoliosis	T10-L4	PSO	L1	265	500	200	N
4	M	22	CS	Kyphoscoliosis	T4-L4	VCRs	T12/L1	455	1000	850	N
5	M	38	TB	Kyphosis	T4-L2	VCRs	T8/T9/T10	370	700	300	N
6	F	34	CS	Scoliosis	T1-T12	PSO	T5	380	1500	1300	N
7	M	33	AS	Kyphosis	T8-L5	PSOs	L1/L3	300	400	200	N
8	F	28	CS	Kyphoscoliosis	T4-L5	VCRs	T10/T11/T12	300	800	1575	N

ADIS, Adult Idiopathic Scoliosis; CS, Congenital Scoliosis; TB, Tuberculosis; AS, Ankylosing Spondylitis; VCR, Vertebral Column Resection; PSO, Pedicle Subtraction Osteotomy; EBL, Estimate Blood Loss.

**Table 2 jpm-13-00602-t002:** Radiological parameters of enrolled patients.

No.	Pre-Operation				Post-Operation				Designed Osteotomy Angle (°)		Achieved Osteotomy Angle (°)		Achieved Correction Angle (°)		ROES (%)		Pedicle Accuracy	Modified Macnab Criteria
	Scoliosis (°)	Kyphosis (°)	Trunk Shift (mm)	SVA (mm)	Scoliosis (°)	Kyphosis (°)	Trunk Shift (mm)	SVA (mm)	Coronal	Sagittal	Coronal	Sagittal	Coronal	Sagittal	Coronal	Sagittal		
1	88	112	15	29	27	54	26	35	43	30	41	33	61	58	95.35	110.0	20/21 (95.24%)	Excellent
2	118	96	36	22	60	34	14	5	48	25	45	23	58	62	93.75	92.0	21/23 (91.30%)	Excellent
3	85	92	10	13	21	8	0	7	35	35	37	33	64	74	105.71	94.29	11/12 (91.67%)	Excellent
4	107	118	19	59	31	45	7	34	60	45	57	47	76	73	95.0	104.44	20/22 (90.91%)	Good
5	-	99	0	0	-	34	0	14	-	50	-	46	-	65	-	92.0	16/16 (100%)	Excellent
6	87	-	32	2	30	-	3	2	35	-	36	-	57	-	102.86	-	17/19 (89.47%)	Excellent
7	-	115	-	620	-	74	-	330	-	30	-	26	-	41	-	86.67	16/16 (100%)	Good
8	111	138	16	35	36	45	28	23	57	55	55	51	75	93	96.49	92.73	18/21 (85.71%)	Excellent

SVA, sagittal vertical axial; ROES, ratio of osteotomy execution and simulation.

## Data Availability

Data is unavailable due to privacy.

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
