# Peer review of "Clinical Application of Personalized Digital Surgical Planning and Precise Execution for Severe and Complex Adult Spinal Deformity Correction Utilizing 3D Printing Techniques"

_jpm, 2023, doi:10.3390/jpm13040602_

Round 1

Reviewer 1 Report

Thank you very much for the opportunity to review interesting research.

I have a question about the time of observation after the procedure: were the analyzed indicators assessed immediately after the procedure or at some time interval? Please clarify.

In addition, in the description of the method, please describe in detail the indicators that were assessed.

Author Response

We thank you sincerely for the scrupulous review and the kind advices on the manuscript. These suggestions are scientific, rigorous and play an important role in improving the quality of our research. In response to the question you raised, we have made serious modifications, and the reply is as follows:

To Q1

ANSWER

The indicators were measured after the patient has completed a post-operative examination, usually about 3 days after surgery. We added some descriptions about the assessing method of the indicators.

Reviewer 2 Report

A case series is presented by the authors describing individualised digital preoperative planning, fabrication of 3D models and intraoperative use of 3D printed guidance aids. This is a descriptive presentation of a case series.  

In principle, such a case series can of course be published, but I would ask you to consider the following points in order to ensure a basic scientific presentation:

1. it is not clear from the text whether the data was collected prospectively or retrospectively, therefore a retrospective compilation is to be assumed. In future publications, please ensure that the methodological structure of your work is carefully described.

All the techniques presented in this case series are neither innovations of existing techniques nor newly established techniques. I therefore find it difficult to understand why it should be published here as a technical note.  Likewise, the term study is not ideal for a case number of 8 patients - the use of the term case series is more appropriate.

3. pictures: Mainly very detailed photos and graphics in small resolution format are used - please consider using more detailed photos in large format. Especially the guides for the osteotomies can then be better understood. 

4. the limitations of the study are not really discussed! Line 395 - 397 is supposed to be a limitation? If they discuss the possibility of further improvement when 97% accuracy has been achieved, this is not a discussion of a limitation... What about the intraoperative use of the 3D models and guidance aids? Reality shows that extremely precise preparations are required to achieve these high accuracies - this is usually associated with more time expenditure... What is new about your study and what is not? 

And please avoid unnecessarily enlarging the discussion text by repeating results that have already been presented in tables or in the results section (e.g. lines 358-368).

5. Small details:

Line 52-56: please rephrase, the sentence is hardly understandable.

Line 176: please correct the word comparison.

Line 247: you describe 8 patients, 3 of them male and 2 female? 3 patients are missing...

6. please ensure that the publication is readable. Sentence lengths over several lines must be grammatically/orthographically perfect, otherwise they are not comprehensible - if this cannot be guaranteed, then it is better to formulate several shorter sentences. 

Author Response

We thank you sincerely for the scrupulous review and the kind advices on the manuscript. These suggestions are scientific, rigorous and play an important role in improving the quality of our research. In response to the question you raised, we have made serious modifications, and the reply is as follows:

To Q1

ANSWER

The data was collected retrospectively in the research. We have added such description in the abstract and method part of the manuscript.

To Q2

ANSWER

There indeed are some similar articles published. We only changed the printing materials and implied the technique to a wide use. According to your suggestion, we revised the research type to case series rather than a technique note.

To Q3

ANSWER

We modified the pictures in the manuscript.

To Q4

ANSWER

We revised the limitation discussion. There are still some technique shortcomings in performing anterior column osteotomy, that is our research direction next.

For the intraoperative use of the 3D models and guidance aids, we have verified the efficacy on spinal surgeries in previous report[1], and found that 3DP spine models can enhance surgeons’ confidence in performing higher grade osteotomies and improve the safety and efficiency in severe spine deformity correction surgery. We think 3DP models may improve the understanding of the anatomically complex sites of skeletal structure, allow surgeons to appreciate the structure and relations of the relevant anatomy much better than the visualization provided by 2-dimensional CT images conventionally used. We have made it a standard procedure in spinal deformity correction surgery now, including 3D model printing and osteotomy guide template printing.

Besides, we hope to ensure the safety of surgery to the greatest extent on the basis of ensuring the accuracy of osteotomy. Before surgery, we will accurately formulate the surgical plan according to the patient-specific anatomical structure. The time Cost and money spent will be cost-effective in the face of postoperative complications or even catastrophic consequences for patients.

As for the result showing in discussion part, we compared the result in our study with other published result. And we have simplified the concerning sentences.

To Q5

ANSWER

We revised the related descriptions you mentioned. That was our mistake.

To Q6

ANSWER

We were aware of the deficiencies in the writing of the manuscript, and we made some changes where possible.

Reference

  1. Pan, A.; Ding, H.; Hai, Y.; Liu, Y.; Hai, J.J.; Yin, P.; Han, B. The Value of Three-Dimensional Printing Spine Model in Severe Spine Deformity Correction Surgery. Global Spine J 2021, 21925682211008830, doi:10.1177/21925682211008830.

Round 2

Reviewer 2 Report

The authors have revised the publication and addressed all points sufficiently. As it stands, I would support publication.